# Inhibition of NHE-1 Increases Smoke-Induced Proliferative Activity of Barrett’s Esophageal Cell Line

**DOI:** 10.3390/ijms221910581

**Published:** 2021-09-30

**Authors:** Eszter Becskeházi, Marietta Margaréta Korsós, Eleonóra Gál, László Tiszlavicz, Zsófia Hoyk, Mária A. Deli, Zoltán Márton Köhler, Anikó Keller-Pintér, Attila Horváth, Kata Csekő, Zsuzsanna Helyes, Péter Hegyi, Viktória Venglovecz

**Affiliations:** 1Department of Pharmacology and Pharmacotherapy, University of Szeged, H-6721 Szeged, Hungary; eszter.becskehazi@gmail.com (E.B.); margaretakorsos@gmail.com (M.M.K.); galeleonora@gmail.com (E.G.); 2Department of Pathology, University of Szeged, H-6725 Szeged, Hungary; tiszlavicz.laszlo@med.u-szeged.hu; 3Biological Research Centre, Institute of Biophysics, H-6726 Szeged, Hungary; hoyk.zsofia@brc.hu (Z.H.); deli.maria@brc.hu (M.A.D.); 4Department of Biochemistry, University of Szeged, H-6720 Szeged, Hungary; kohler.zoltan.marton@gmail.com (Z.M.K.); keller.aniko@med.u-szeged.hu (A.K.-P.); 5Department of Pharmacognosy, University of Szeged, H-6720 Szeged, Hungary; horvath.attila@pharmacognosy.hu; 6Department of Pharmacology and Pharmacotherapy, Medical School & Szentágothai Research Centre, University of Pécs, H-7624 Pécs, Hungary; csekoe.kata@gmail.com (K.C.); zsuzsanna.helyes@aok.pte.hu (Z.H.); 7PharmInVivo Ltd., H-7629 Pécs, Hungary; 8First Department of Medicine, University of Szeged, H-6720 Szeged, Hungary; hegyi.peter@pte.hu; 9Medical School & Szentágothai Research Centre, Institute for Translational Medicine, University of Pécs, H-7624 Pécs, Hungary; 10Division of Gastroenterology, First Department of Medicine, Medical School, University of Pécs, H-7624 Pécs, Hungary

**Keywords:** esophagus, ion transport, smoking, NHE-1, Barrett’s esophagus

## Abstract

Several clinical studies indicate that smoking predisposes its consumers to esophageal inflammatory and malignant diseases, but the cellular mechanism is not clear. Ion transporters protect esophageal epithelial cells by maintaining intracellular pH at normal levels. In this study, we hypothesized that smoking affects the function of ion transporters, thus playing a role in the development of smoking-induced esophageal diseases. Esophageal cell lines were treated with cigarettesmoke extract (CSE), and the viability and proliferation of the cells, as well as the activity, mRNA and protein expression of the Na^+^/H^+^ exchanger-1 (NHE-1), were studied. NHE-1 expression was also investigated in human samples. For chronic treatment, guinea pigs were exposed to tobacco smoke, and NHE-1 activity was measured. Silencing of NHE-1 was performed by using specific siRNA. CSE treatment increased the activity and protein expression of NHE-1 in the metaplastic cells and decreased the rate of proliferation in a NHE-1-dependent manner. In contrast, CSE increased the proliferation of dysplastic cells independently of NHE-1. In the normal cells, the expression and activity of NHE-1 decreased due to in vitro and in vivo smoke exposure. Smoking enhances the function of NHE-1 in Barrett’s esophagus, and this is presumably a compensatory mechanism against this toxic agent.

## 1. Introduction

Cigarette smoking is responsible for the development of many diseases, especially different types of cancers. Since smoking primarily affects the lungs, the effects of smo-king have been most intensively studied on this organ. However, other organs may also be affected, such as the esophagus, which is directly exposed to cigarette smoke. For this reason, a number of clinical studies have been conducted to examine the connection bet-ween smoking and esophageal diseases. These studies have shown that smoking strongly correlates with the development of esophageal adenocarcinoma (EAC) and Barrett’s esophagus (BE) and also increases the risk of progression from BE to EAC [1,2,3,4,5]. In contrast, only a few data are available regarding the cellular mechanism of smoking-induced lesions. In an older study, Orlando et al. showed that the esophageal potential difference is reduced by cigarette smoke extract (CSE), in which inhibition of Na^+^ transport plays an important role [6]. This study suggests that smoking alters the ion transport processes in esophageal epithelial cells (EECs); however, it is not known whether this takes part in the development of BE or EAC. 

Ion transport processes through the esophageal mucosa play an important protective role, as they greatly contribute to the maintenance of normal intracellular pH (pH_i_). Se-veral ion transporters have been identified on EECs in recent years, and their role has been characterized both under physiological and pathophysiological conditions [7,8]. Our workgroup showed that acid and/or bile acids alters the activity and expression of ion transporters, which may be important in the development and progression of esophageal diseases [8]. Among the acid–base transporters, the Na^+^/H^+^ exchanger (NHE) is one of the most important transmembrane protein that mediates the exchange of Na^+^ and H^+^. In addition to playing an important role in the alkalization of the pH_i_, it also regulates cell vo-lume, proliferation, migration, and invasion [9,10,11]. Several members of the NHE family are known, of which NHE-1 is the most common, ubiquitously expressed isoform. The presence of NHE-1 has been shown in the esophagus of several species, such as rat and rabbit, where it is essentially involved in the regulation of pH_i_ [12]. In contrast, the expression of this isoform in the normal human esophagus is controversial, and its importance is more highlighted under pathological conditions [8,13,14,15,16,17]. Increased NHE-1 expression has been shown in BE, which probably plays a protective role against the acid- or bile-induced injury by enhancing the cellular resistance of the cells [8,15,18]. The role of NHE-1 in EAC is controversial. Some studies suggest that NHE-1 enhances the growth of eso-phageal cancer cells, while other studies have shown that NHE-1 expression is associated with longer postoperative survival [13,16].

There is no information on how smoking affects NHE-1 activity or expression in the esophagus. Since Na^+^ transport is inhibited by smoking [6], it is conceivable that NHE-1 plays a role in the pathogenesis of cigarette-smoke-induced esophageal diseases. Therefore, the objective of the present study was to investigate the effect of tobacco smoke on normal, metaplastic and dysplastic cells and to investigate the role of NHE-1 in it.

## 2. Results

### 2.1. Effect of CSE on Esophageal Epithelial Cell Proliferation

To examine the effect of CSE on cell proliferation, first we determined the concentrations of CSE at which the cells retained their viability. CSE concentrations were chosen based on the literature data [19,20]. Cytotoxicity studies showed that both CP-A and CP-D cells mostly tolerated CSE at 1 and 10 µg/mL concentrations, at each incubation time (6, 24 and 72 h). In contrast, 100 µg/mL CSE induced a high degree of cell death, especially during longer incubation (Figure 1a). Therefore, in the proliferation assays, the effect of 1 and 10 µg/mL CSE was examined for 6, 24 and 72 h (Figure 1b). In the metaplastic, CP-A cells, CSE treatment dose-dependently reduced cell proliferation in the 24 and 72 h treatment groups. In contrast, in the dysplastic, CP-D cells 72 h CSE treatment significantly increased the proliferation. 

### 2.2. Activity and Expression of NHE-1 in the Metaplastic and Dysplastic Cells

Next, we examined the rate of NHE activity in the CP-A and CP-D cell lines, using the NH_4_Cl pre-pulse technique (Figure 2a,b). As shown in Figure 2a, in the absence of HCO_3_^-^ the initial rate of regeneration from acidosis reflects the activity of NHE. Currently, nine NHE isoforms are known, and among them the presence of NHE-1 and NHE-2 was confirmed in the esophageal mucosa [8,12,21]. Our previous studies showed that NHE-1 displays greater activity and is better expressed than NHE-2 both in the metaplastic and dysplastic cells [8], indicating that NHE-1 is primarily responsible for the regeneration from acidosis. As shown in Figure 2a,b, regeneration from acidosis was higher in CP-A (BF: 5.47 ± 0.52) than in CP-D cells (BF: 3.36 ± 0.24), indicating that CP-A cells have higher NHE-1 activity. We have also compared the mRNA and protein expressions of NHE-1 between the metaplastic and dysplastic cells at different time points (6, 24 and 72 h). In addition, mRNA expression of NHE-1 (*SLC9A1*) was investigated by RT-PCR 24 h after plating the cells. As an internal gene, human beta actin (*ACTB*) was used. RT-PCR analysis revealed that there was no significant difference in NHE-1 expression among CP-A and CP-D cells, and no difference was observed between the different incubation times (Figure 2c). Similar to RT-PCR, the Western blot analysis showed no difference in the protein expression of NHE-1 between the CP-A and CP-D cells (Figure 2d).

### 2.3. Effect of CSE on The Activity and Expression of NHE-1 

In order to investigate the effect of CSE on the activity of NHE-1, the previously mentioned NH_4_Cl pre-pulse technique was used. Since CSE alone can affect the fluorescence signals, cells were pretreated with 1, 10 or 100 µg/mL CSE for 1 h, and then NHE activity was measured (Figure 3a,b). Control cells were incubated in HEPES solution without CSE. In the case of CP-A cells, pretreatment with 1 µg/mL CSE decreased NHE activity from 5.47 ± 0.52 to 3.08 ± 0.55. In contrast, at higher concentrations (10 and 100 µg/mL, respectively) the activity of the exchanger increased (8.18 ± 1.3 at 10 µg/mL CSE and 12.28 ± 0.73 at 100 µg/mL CSE). In CP-D cells, CSE strongly reduced NHE-1 activity at all three concentrations (from 3.36 ± 0.24 to 1.25 ± 0.26 at 1 µg/mL CSE, 1.62 ± 0.23 at 10 µg/mL CSE and 1.46 ± 0.29 at 100 µg/mL CSE, respectively). 

The CSE treatment did not cause significant differences in mRNA expression of NHE-1 in any of the cell lines (Figure 4a). In contrast, the protein expression was increased in the CP-A cells upon CSE treatment, almost in all treated groups (Figure 4b). In the CP-D cells, CSE treatment caused a robust increase at 1 µg/mL concentration in the 6 h treatment group, while no significant change was detected in the other groups. 

### 2.4. Smoking Decreases NHE-1 Activity on Normal Esophageal Epithelial Cells

In order to investigate how CSE affects NHE activity under physiological conditions, we studied the effect of CSE on normal EECs isolated from guinea pigs. The same concentrations were used as for the cell lines, and the cells were pretreated with CSE in the same manner. As shown in Figure 5a,b, NHE activity was significantly reduced by CSE treatment (from 12.19 ± 0.46 to 4.64 ± 0.94 at 1 µg/mL CSE, to 3.96 ± 0.43 at 10 µg/mL CSE and to 4.49 ± 0.4 at 100 µg/mL CSE, respectively). In order to investigate the chronic effects of smoking, guinea pigs were exposed to cigarette smoke for one, two and four months, respectively, and then NHE activity was examined (Figure 5c). Guinea pigs of the same age were used as controls. Similar to acute CSE treatment, chronic treatment decreased NHE activity from 15.81 ± 0.91 to 7.98 ± 0.52 in the 1-month group, from 9.92 ± 0.78 to 7.9 ± 0.33 in the 2-month group and from 10.86 ± 0.54 to 5.46 ± 0.19 in the 4-month group (Figure 5d). These data indicate that smoking decreases the activity of NHE-1 in the normal eso-phageal mucosa. 

### 2.5. Effect of Smoking on NHE-1 Protein Expression in Human Esophageal Samples

Protein expression of NHE-1 was investigated in normal squamous epithelium and in BE samples obtained from patients with smoking and non-smoking history (Figure 6a,b). Patients who had never smoked or not smoked for more than a year were classified as non-smokers, while patients who had been smokers for at least 20 years were classified as smokers. Only patients with known smoking status were included in the analysis. As controls, normal esophageal biopsy samples and the intact tumor-free margin of surgically resected esophageal cancer were used. Weak NHE-1 expression was detected in the normal esophageal epithelium, and it was further reduced by smoking. In BE, strong NHE-1 expression was observed, mainly at the basolateral membrane of the columnar cells. In smokers, NHE-1 expression increased, and staining was detected not only in the plasma membrane but in the cytoplasm as well. Interestingly, strong NHE-1 staining was also observed in the glands. There was no significant difference between the intestinal and non-intestinal metaplasia, neither in the smoker nor in the non-smoker group. 

### 2.6. Role of NHE-1 in The CSE-Induced Proliferation

In order to investigate whether the altered expression or activity of NHE-1 has any role in the effect of CSE on proliferation, we silenced the *SLC9A1* gene, using specific siRNA (Figure 7a–d). The efficiency of silencing was investigated at both mRNA and protein levels (Figure 7a,b). In CP-A cells, NHE-1 knockdown reduced the rate of proliferation at each incubation time, suggesting that NHE-1 is essential for the normal function of the cells (Figure 7c). In the CP-D cells, the lack of NHE-1 protein initially increased the rate of proliferation, whereas no significant difference was observed with additional incubation times (Figure 7c). In the absence of NHE-1, CSE treatment increased the rate of proliferation in the CP-A cells in almost all treated groups. For CP-D cells, proliferation increased alone in the 72h treatment group (Figure 7d). 

## 3. Discussion

Recent studies have described that altered expression or activity of ion transporters play an important role in the development or progression of different types of cancer [22,23,24,25,26]. For this reason, many ion transporters emerged as potential targets for cancer therapy [27,28]. Our study demonstrates, for the first time, that smoking affects cell proliferation in BE in which the ubiquitously expressed transmembrane transporter, NHE-1, plays a central role. In the presence of NHE-1, CSE decreased the proliferation of metaplastic cells, whereas, in the absence of the exchanger, cell proliferation increased due to the CSE treatment. This result may be significant from the point of view that NHE-1 plays a protective role in BE, and decreased NHE-1 expression may contribute to the neoplastic progression of BE in smoking patients. 

An interesting observation of our study is that CSE treatment slightly reduced the proliferation of metaplastic cells, while it increased the proliferation of dysplastic cells. The decreased proliferation in the metaplastic cells is presumably due to the decreased cell viability at higher concentrations of CSE. In contrast, dysplastic cells were much more resistant to CSE, and despite the low degree of cell death, cell proliferation increased with treatment. In order to study the underlying mechanisms, we investigated the effect of smoking on ion transport processes. Esophageal ion transporters play an important protective role in EECs by preventing acidic or basic shift in pH_i_. Therefore, disruption of pH regulatory processes leads to an upset of extra- and intracellular pH, which can result in changes in cellular function and also causes genetic instability. It is well-known that the pH of tumor is dysregulated and typically an acidic microenvironment develops within the tumor that promotes cell division and migration [29,30,31,32]. Among the ion transporters NHE-1 is an ubiquitously expressed plasma membrane protein that plays an essential role in maintaining physiological pH_i_. Inadequate function of this transporter has been described in several cancer types, including esophageal cancer [13,16,33,34,35,36,37]; therefore, NHE-1 emerged as a potential target in anti-cancer therapy [37,38]. We showed that, in the case of acute CSE exposure, higher concentrations of CSE increased NHE-1 activity in the metaplastic cells; this is presumably a defense mechanism by which the cells try to maintain the normal pH homeostasis. In contrast, decreased activity of NHE-1 in CP-D cells indicates damaged compensatory pH regulatory mechanisms. We have previously shown that bile, which is an important etiological factor in the development of GERD and BE, had the opposite effect that is decreased the activity of NHE-1 in the metaplastic cells and increased it in the dysplastic cells [8]. In order to get a more complete picture of the effect of CSE, we treated the cells with CSE for 6, 24 and 72 h and the mRNA, and the protein expressions of NHE-1 were investigated. In CP-A cells, CSE treatment induced an increase in mRNA expression, but this did not reach a significant level. However, a clear elevation was found at the protein level that is thought to be associated with the increased NHE-1 activity by CSE. Examination of human esophageal samples also showed that smoking increases NHE-1 expression in both intestinal and non-intestinal metaplasia, consistent with the results obtained on the CP-A cells. In the case of CP-D cells, no signi-ficant change in either mRNA or protein expressions was observed after long-term or higher-dose CSE administration, indicating that only the activity of the protein changes due to the CSE treatment. In order to investigate how smoking affects NHE activity under normal conditions, we examined the effect of acute and chronic tobacco exposure on guinea pig EECs. We chose guinea pigs because we have previously shown that ion-transport processes in the secretory gland of the guinea pig is similar to humans; in addition, more cells can be obtained from the esophagus of guinea pig than from the esophagus of mice or rat [39]. The duration of chronic in vivo smoking was determined based on the literature data [40,41,42]. Our results showed that both acute and chronic tobacco smoke exposures significantly reduce NHE-1 function. In the case of acute CSE exposure no dose-dependent effect was observed. This can be explained by the fact, that the composition of each CSE preparation may slightly differ from each other as it contains thousands of components, most of which are unstable molecules. Since the CSE extract was not analyzed and only the concentration of the whole extract was calculated minor or larger differences in the preparations may be responsible for the lack of a dose-response effect. In the case of chronic smoking, smoking did not cause as much a decrease in the 2-month-old guinea pigs as in the 1- and 4-month-old animals. One explanation for this is that the effects of smoking and/or the activity of NHE-1 differs in each age group. The effect of chronic smoking in humans was only studied at expression level because human EECs are not suitable for functional measurements due to the high sensitivity and low viability of the human primary cells. Consistent with the result obtained in guinea pig EECs, smoking reduced the expression of NHE-1 in humans. Although NHE-1 is expressed in a very low level in the normal human esophageal mucosa [8,13,15,16], the decreased expression of NHE-1 associates with cellular acidosis, which may increase the risk of cancer development, as a greater number of DNA damage and thus mutations develop in an acidic environment [15]. In order to clarify the role of NHE-1 in the CSE-induced proliferation, we downregulated NHE-1 by specific siRNA transfection. In the absence of NHE-1, the CSE-induced proliferation increased in the metaplastic cell line, suggesting that NHE-1, in addition to being essential in maintaining the normal pH of cells, also performs an important protective function and regulates cell proliferation against toxic agents. The protective role of NHE-1 against the carcinogenic processes has been also demonstrated in esophageal squamous cell carcinoma (ESCC), where suppression of NHE-1 increased the malignant potential and associated with poor prognosis in ESCC patients [13]. In contrast, Guan et al. have found that inhibition of NHE-1 suppressed esophageal cancer cell growth in EAC and ESCC cell lines and in nude mouse xenografts [16]. It has also been demonstrated in other cancer types that NHE-1 promotes tumor malignancy by providing appropriate pH conditions for tumor growth and invasion [11,43]. These data suggest that NHE-1 acts as a tumor oncogene rather than a suppressor in cancer. In BE, most studies agree that increased NHE activity and/or expression is more likely part of a defense or adaptive mechanism which protects cells against toxic-agent-induced cellular acidification [8,14,15,18,44]. In the more advanced dysplastic state, inhibition of NHE-1 had no effect on the CSE-induced proliferation, indicating that, in dysplasia, the proliferative effect of CSE is independent from NHE-1.

Taken together, smoking affects NHE-1 function in normal, metaplastic and dysplastic cells differently. Under normal conditions, smoking reduces the activity and expression of NHE-1, resulting in the acidosis of pH_i_. Disturbance of the pH homeostasis can lead to cell deaths or to the malignant transformation of the cells. In the metaplastic state, smoking increases the function of NHE-1, which is presumably a compensatory mechanism that prevents the onset of cancerous processes by keeping the intracellular pH in the normal range. As the expression of NHE-1 decreases, this protective mechanism disappears and the proliferative potential of the cells increases. In contrast to BE, decreased activity or expression of NHE-1 had no effect on smoking-induced proliferation in the dysplastic state indicating the involvement of other mechanisms.

We propose that upregulation of NHE-1 is the part of a protective mechanism against the harmful effects of smoking; however, further investigation would be needed to support this hypothesis. Direct increase in NHE-1 expression by using NHE-1 agonists or the use of transgenic mice models in which the *SLC9A1* gene is modified would give a more complete picture of the role of NHE-1. Nevertheless, the present results indicate that direct augmentation of NHE-1 function may provide new avenues for decreasing the damaging effect of smoking. 

## 4. Materials and Methods

### 4.1. Chemicals and Solutions

All general laboratory chemicals and Trypan Blue solution (Catalogue No. T8154) were purchased from Sigma-Aldrich (Budapest, Hungary). We obtained 2,7-bis-(2-carboxyethyl)-5 (6)-carboxyfluorescein acetoxymethyl ester (BCECF-AM) from Invitrogen (Watham, MA, USA), prepared (2 µmol/L) in dimethyl sulfoxide and stored at −20 °C. Nigericin (10 µM) was dissolved in ethanol and stored at −20 °C. High-Capacity Reverse Transcription Kit, TaqMan gene expression Master Mix, lipofectamine 2000 transfection reagent were purchased from Thermo Fisher Scientifc (Watham, MA, USA). Cell Counting Kit-8 was from Dojindo Molecular Technologies (Rockville, MD, USA). Citotoxicity detection kit (LDH) was obtained from Roche (Catalogue No. 11644793001, Roche). *SLC9A1* siRNAs (siRNA ID: 119643) was from Life Technologies. The standard Na^+^-HEPES solution contained (in mM): 130 NaCl, 5 KCl, 1 CaCl_2_, 1 MgCl_2_, 10 D-glucose and 10 Na-HEPES. NH_4_Cl-HEPES solution was supplemented with 20 mM NH_4_Cl, while NaCl concentration was lowered to 110 mM. HEPES-buffered solutions were gassed with 100% O_2_, and their pH was set to 7.5 with HCl. 

### 4.2. Animals

Guinea pigs (4–12 weeks old, male) were kept in standard plastic cages on 12:12 h light–dark cycle at room temperature (23 ± 1 °C) and had free access to standard laboratory chow and drinking solutions. Animal experiments were conducted in accordance with the *Guide for the Care and Use of Laboratory Animals* (United States, Department of Health and Human Services). In addition, the experimental protocol performed on non-smoking animals was approved by the local Ethical Board of the University of Szeged, Hungary.

### 4.3. Patients

Formalin-fixed paraffin-embedded esophageal tissue samples were retrieved from the archives of the Department of Pathology, University of Szeged (Szeged, Hungary), from January 2018 to May 2021. Retrospective data collection of patients was performed by the approval of the Ethics Committee of the University of Szeged (No. 4658), according to Helsinki Declaration and GDPR. We collected data based on diagnosis, histopatholo-gical features of the esophageal lesion, state of the disease and smoking history of the patients from patient database used in Hungarian healthcare system (eMedSolution). Each esophageal biopsy and surgical resection sample was analyzed by pathologists earlier. Patients were basically classified into two groups: smoking group with a smoking history of more than 20 years and non-smoking group who had never smoked or had not smoked for at least a year. As a control, tumor-free resection margins and normal esophageal biopsy samples were used. The average age of BE patients was 55.1 ± 4.6 years in the smoking group (*n* = 7), and 57.3 ± 3.8 years in the non-smoking group (*n* = 20). The average age in the control group was 38.5 ± 11.5 years in the smoking group (*n* = 3), and 59.7 ± 3.8 years in the non-smoking group (*n* = 7).

### 4.4. Cell Cultures

CP-A (human metaplastic esophageal epithelial cell line) and CP-D (human dysplastic esophageal epithelial cell line) cells were purchased from American Type Culture Collection (Manassas, VA, USA). The complete growth medium consists of MCDB-153 basal medium, 5% fetal bovine serum, 4 mM L-glutamine, 1× ITS supplement (Sigma I1884; 5 µg/mL Insulin, 5 µg/mL Transferrin and 5 ng/mL Sodium Selenite), 140 µg/mL Bovine Pituitary Extract (Sigma P1476), 20 mg/L adenine, 0,4 µg/mL hydrocortisone, 8,4 µg/L cholera toxin (Sigma C8052), 20 ng/mL recombinant human EGF (Epidermal Growth Factor) and 1% (*v*/*v*) penicillin/streptomycin. The cells were cultured at 37 °C and gassed with a mixture of 5% CO_2_–95% air. Cells were seeded at 100% confluency and were used between 3 and 19 passage numbers.

### 4.5. Preparation of Cigarette-Smoke Extract

CSE was prepared at the Department of Pharmacognosy, University of Szeged. Briefly, mainstream smoke from 15 Kentucky Research Cigarettes (3R4F; 12 mg tar and 1.0 mg nicotine/cigarette), was continuously bubbled through 10 mL of distilled water. Dry weight was measured after evaporation of the crude extract with N_2_. CSE solution was then diluted to the appropriate concentration, using HEPES or culture media. CSE was freshly prepared for each experiments or used within 2 days of preparation.

### 4.6. Cigarette-Smoke Exposure

The chronic effect of cigarette smoking was investigated by using a smoking chamber at the Department of Pharmacology and Pharmacotherapy, University of Pécs. Male guinea pigs were divided into three groups, according to the period of cigarette-smoke exposure (1-, 2- and 4-month exposure, *n* = 3/group). In order to avoid large age differrences at the time of sacrifice, animals were selected for each group based on their age, so all animals were 5 months old at the time of sacrifice. Guinea pigs were maintained under 12:12 h light/dark cycle, with free access to food and water. Animals were exposed to whole body cigarette-smoke exposure 4 times a day, 5 days a week, for 30 min each time, using a TE2 whole body smoke exposure chamber (Teague Enterprises, Woodland, CA, USA). During the experiment, 3R4F Kentucky Research Cigarettes (Kentucky Tobacco Research and Development Center, Lexington, KY, USA) were smoked and ventilated inside the chamber. The animals were sacrificed 24–48 h after the last CSE exposure, and EECs were isolated. Intact age- and sex-matched animals served as controls. All experimental procedures were in accordance with the institutional guidelines under approved protocols (No. XII./2222/2018, University of Pécs).

### 4.7. Isolation of Guinea Pig Esophageal Epithelial Cells

Animals were sacrificed by cervical dislocation, the esophagus was removed and EECs were isolated as described previously [45]. Briefly, the organ was cut longitudinally, rinsed in Hank’s Balanced Salt Solution (HBSS, Sigma H9269) and digested in dispase solution (2 U/mL, Sigma D4818) for 40 min. After digestion, the inner, epithelial layer of the esophagus was detached from the submucosa and rinsed in HBSS. Then the epithelial layer was incubated in 0.5% Trypsin–EDTA solution supplemented with 1% (*v/v*) antimycoticum/antibioticum for 2 × 15 min. The trypsin was inhibited by a filtered Soybean trypsin inhibitor (Gibco, 10684033) solution, and the whole lysate was centrifuged for 5 min, at 1000 rpm. The cell pellet was diluted in keratinocyte serum free media (KSFM, Gibco, Catalogue No. 17005042) supplemented with 1% (*v/v*) antimycoticum/antibioticum and seeded onto cover glasses and incubated until use. Viability of guinea pig EECs was investigated by using Trypan Blue reagent. After the incubation, bright field images were taken under 40× magnification, and stained cells were counted and considered not viable. 

### 4.8. Immunohistochemistry 

Immunohistochemical analysis of NHE-1 expressions was performed on 4% buffered formalin-fixed sections of human esophageal samples embedded in paraffin. The 5 µm–thick sections were stained in an automated system (Autostain, Dako, Glostrup, Denmark). Briefly, the slides were deparaffinized, and endogenous peroxidase activity was blocked by incubation with 3% H_2_O_2_ (10 min). Antigenic sites were disclosed by applying citrate buffer in a pressure cooker (120 °C, 3 min). To minimize non-specific background staining, the sections were then pre-incubated with milk (30 min). Subsequently, the sections were incubated with a human anti-NHE-1 (1:100 dilution, 30 min, Alomone Laboratories) primary polyclonal antibody and exposed to LSAB2 labeling (Dako, Glostrup, Denmark) for 2 × 10 min. The immunoreactivity was visualized with 3,3′-diaminobenzidine (10 min); then the sections were dehydrated, mounted and examined. NHE-1 expressing cells were identified by the presence of a dark red/brown chromogen. A semi-quantitative scoring system was used to evaluate the expression of NHE-1. The intensity of staining (0 = negative, 1 = weak, 2 = moderate and 3 = strong) and the percentage of positive cells (1–0–25% of the cells are positive, 2–25–50% of the cells are positive, 3–50–75% of the cells are positive and 75–100% of the cells are positive) were scored and then the composite score was obtained by multiplying the intensity of staining and the percen-tage of immunoreactive cells. 

### 4.9. Quantitative Real-Time PCR Analysis

Total mRNA was isolated by using an RNA isolation kit of Macherey-Nagel (Nucleospin RNA Plus kit, Macherey-Nagel, Germany) according to manufacturer’s instructions. The concentration of RNA was determined by spectrophotometry (NanoDrop 3.1.0, Rockland, DE, USA). Two micrograms of total RNA were reverse-transcribed, using High-Capacity cDNA Archive Kit (Applied Biosystems) according to manufacturer’s instructions. Quantitative real-time PCR was carried out on a Roche LightCycler 96 SW (Roche, Basel, Switzerland). TaqMan probe sets of *SLC9A1* were used to check gene expression. Target gene expression levels were normalized to a human housekeeping gene, β-actin (*ACTB*), and then, using the ΔΔC_T_ method, relative gene expression was calculated. Fold changes were represented (2^−ΔΔCT^). Values below 0.5 and above 2.00 were considered significant.

### 4.10. Western Blot

Cells were lysed in Cell Lysis Buffer (Catalogue No. 9803, Cell Signaling Technology, Danvers, MA, USA) supplemented with complete EDTA-free protease inhibitor (Roche, Catalogue No. 11873580001). Then samples were centrifuged at 2500 rpm for 20 min at 4 °C, and the supernatants were used. Protein concentration in the samples was determined by using a BCA assay (Pierce Chemical, Rockford, IL, USA) or Bradford reagent (Bio-Rad Laboratories, Hercules, CA, USA), and equal amounts of proteins (20 or 30 µg) were resolved on polyacrylamide gel and transferred onto Protran (GE Healthcare Amersham™) or PVDF (Invitrogen, Watham, MA, USA) membranes. Membranes were incubated overnight with rabbit polyclonal anti-NHE-1 (Catalogue No. ANX-010, Alomone Labs, Jerusalem, Israel), mouse monoclonal anti-GAPDH (Catalogue No. MAB 374, Sigma Aldrich, Hungary) or mouse monoclonal anti-α-Tubulin antibody (Catalogue No. T9026, Merck, Darmstadt, Germany) followed by the incubation with the appropriate HRP-conjugated secondary antibody (Catalogue No. P0448 goat anti-rabbit and P0161 rabbit anti-mouse, DAKO, Glostrup, Denmark or G-21040 goat anti-mouse, Invitrogen, Watham, MA, USA). The peroxidase activity was visualized by using the enhanced chemi-luminescence assay (Advansta, Menlo Park, CA, USA) or with Clarity Chemiluminescence Substrate (Bio-Rad Laboratories, Hercules, CA, USA). Signal intensities were quantified by using the QuantityOne software (Bio-Rad, Hercules, CA, USA) or Image Lab Software, version 5.2 (Bio-Rad Laboratories, Hercules, CA, USA). The results from each membrane were normalized to the GAPDH or α-Tubulin values and compared to the 6 h control.

### 4.11. Measurement of Intracellular pH

Cells were seeded onto 24 mm cover glasses which were placed on the stage of an inverted microscope connected with an Xcellence imaging system (Olympus, Budapest, Hungary). Cells were incubated with a pH-sensitive fluorescence dye, BCECF-AM for 30–60 min according to cell type. Cells were perfused with solutions at 37 °C at a 5 to 6 mL/min perfusion rate. Average 5–12 regions of interest (ROIs) were marked in each measurement, and one image was taken per second. The cells were excited with 440 and 495 nm wavelength, and a 440/495 ratio was detected at 535 nm. One pH_i_ measurement was obtained per second. In situ calibration of the fluorescence signal was performed by using the high K^+^-nigericin technique. Since CSE alone influences fluorescence signals, cells were pretreated with freshly prepared CSE (1, 10 and 100 µg/mL) for 1 h before microfluorometric measurements.

### 4.12. Determination of Buffering Capacity

The total buffering capacity (β_total_) of cells was estimated according to the NH_4_Cl pre-pulse technique, as previously described [46,47]. Briefly, EECs were exposed to various concentrations of NH_4_Cl in Na^+^- and HCO_3_^−^-free solutions. The total buffering capacity of the cells was calculated by using the following equation: β_total_ = β_i_ + β_HCO3−_ = β_i_ + 2.3 × [HCO_3_^−^]_i_, where β_i_ refers to the ability of intrinsic cellular components to buffer changes of pH_i_ and was estimated by the Henderson–Hasselbach equation. The measured rates of pH_i_ change (∆pH/∆t) were converted to transmembrane base flux *J*(B^−^), using the following equation: *J*(B^−^) = ∆pH/∆t × β_i_. The β_i_ value at the start point pH_i_ was used for the calculation of *J*(B^−^). 

### 4.13. Measurement of Na^+^/H^+^ Exchanger Activity

For evaluating the activity of NHE-1, NH_4_Cl pre-pulse technique was used. EECs were exposed to NH_4_Cl (20 mM) for 3 min, which resulted in a sudden pH_i_ elevation through NH_3_ diffusion into the cells. NH_4_Cl withdrawal caused a remarkable decrease in pH_i_ as the intracellular NH_4_^+^ and H^+^ dissociating and basic NH_3_ exiting the cells. The regeneration from acidosis (the first 60 s) reflects the activity of NHEs in standard HEPES-buffered solutions. The following equation was used for estimating the transmembrane base flux: *J*(B^−^) = ΔpH/Δt × β_i_, where ΔpH/Δt was calculated by linear regression analysis, whereas the intrinsic buffering capacity (β_i_) was determined by the Henderson–Hasselbach equation.

### 4.14. SLC9A1 Gene Silencing

Cells were seeded on a 6-well plate in antibiotic-free complete growth medium and incubated overnight. *SLC9A1* gene silencing was performed at 40–50% confluency. Then 100 pmol *SLC9A1* siRNA was dissolved in 250 µL Opti-MEM (Gibco, Catalogue No. 31985070) reduced serum medium. Then 5 or 7.5 µL Lipofectamine 2000 was added to 250 µL Opti-MEM and incubated for 5 min at room temperature. Then the prepared siRNA solution and Lipofectamine 2000 were mixed and incubated for 20 min to form complexes. The complexes were added to the wells, mixed gently by rocking the plate back and forth and then incubated for 72 h. After transfection, RT-qPCR and immunocytochemistry was performed to estimate mRNA and protein levels.

### 4.15. Proliferation

Cells were seeded at 10^3^ cells per well into a 96-well plate (100 µL/well) in complete growth medium and allowed to attach for 24 h. Cells were then treated with CSE (1 and 10 µg/mL) for 6, 24 and 72 h. After the treatments, 10 µL of CCK8 solution was added to each well and the cells were incubated for further 3 h. Absorbance was detected at 450 nm, using a FLUOstar OPTIMA Spectrophotometer (BMG Labtech, Ortenberg, Germany). 

### 4.16. Cytotoxicity Assay

For cytotoxicity assay, 100 µL of cell suspension was seeded into a 96-well plate (2.5 × 10^4^ cells/well) and allowed to adhere overnight. On the following day, the cells were incubated with CSE (1, 10 and 100 µg/mL) for 6, 24 and 72 h. Then 100 µL of supernatant from each of the wells was carefully transferred into a new 96-well plate containing 100 µL reaction mixture. We then measured lactate dehydrogenase (LDH) activity at 490 nm using a FLUOstar OPTIMA Spectrophotometer (BMG Labtech, Ortenberg, Germany). For background controls, we measured 200 µL assay medium, without cells. For low controls, we used 100 µL cell suspension and 100 µL assay medium. In the case of high controls, the mixture of 100 µL cell suspension and 100 µL Triton-X 100 (0.1%) solution was measured. The LDH release induced by Triton-X 100 was assigned to 100%. The average absorbance values of each of the triplicates were calculated, and the average value of the background control (LDH activity contained in the assay medium) was subtracted from each of the samples to reduce background noises. We then calculated the percentage of cytotoxicity by using the following formula: Cytotoxicity (%) = (exp. value–low control/high control–low control) × 100. Low control determines the LDH activity released from the untreated normal cells (spontaneous LDH release), whereas high control determines the maximum releasable LDH activity in the cells (maximum LDH release). 

### 4.17. Statistical Analysis

Results were described as means ± SE. For statistical analysis, one-way ANOVA and Student’s *t*-test were used, *p* ≤ 0.05 were considered significant.

## Figures and Tables

**Figure 1 ijms-22-10581-f001:**
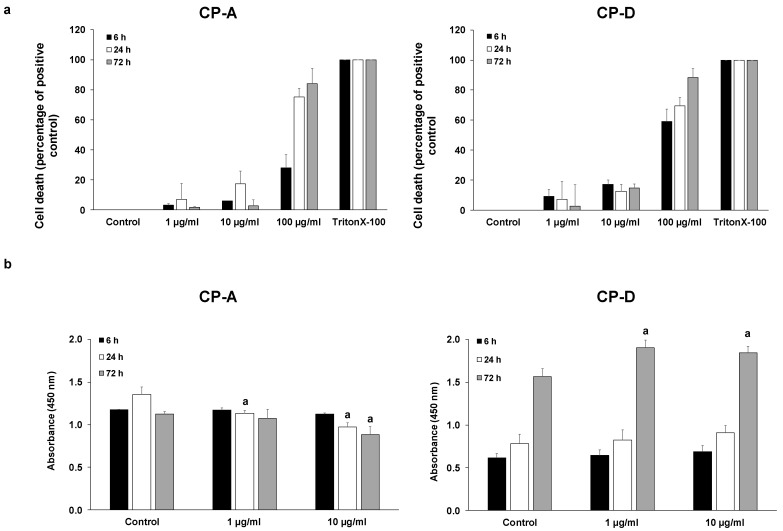
Effects of cigarette smoke extract (CSE) treatment on cell viability and proliferation. Metaplastic (CP-A) and dysplastic (CP-D) esophageal cell lines were exposed to different concentrations of CSE for 6, 24 and 72 h and the effects on cellular viability (**a**) and proliferation (**b**) were studied, using LDH and CCK8 assays, respectively. In the case of viability assay, 0.1% Triton X-100 was used as a positive control. Data represent mean ± SEM of three independent experiments; a = *p* ≤ 0.05 vs. control.

**Figure 2 ijms-22-10581-f002:**
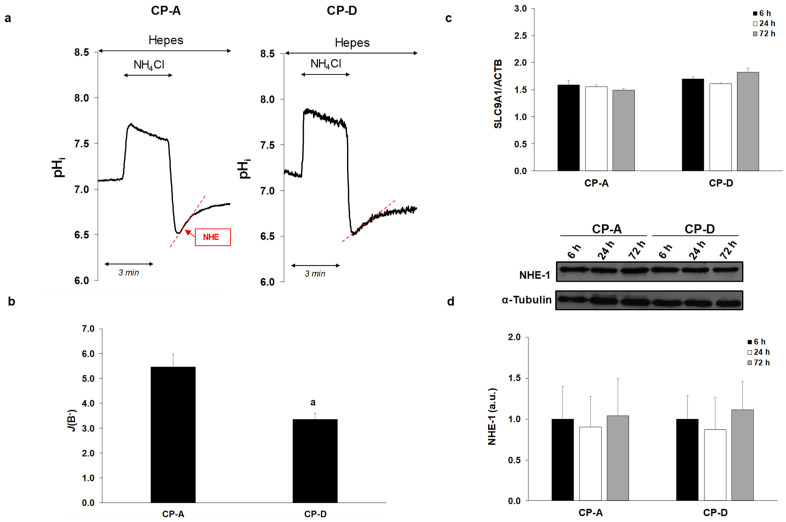
Activity, mRNA and protein expression of Na^+^/H^+^ exchanger-1 (NHE-1) in esophageal cell lines. (**a**) Representative intracellular pH (pH_i_) curves present the recovery from acidosis in CP-A and CP-D cells. (**b**) Summary data of the calculated activity of NHE-1 in the different cell lines. The rate of pH recovery (*J*(B^−^)) was calculated from the ΔpH/Δt obtained via linear regression analysis of the pH_i_ measurement performed over the first 60 s of recovery from the lowest pH_i_ level (initial pH_i_). The buffering capacity at the initial pH_i_ was used to calculate *J*(B^−^). Data are presented as the mean ± SEM. a: *p* ≤ 0.05 vs. CP-A; *n* = 5–11 exp./26–91 region of interests (ROIs). (**c**) mRNA and (**d**) protein expression of NHE-1 in the CP-A and CP-D cells. α-Tubulin was used as a protein-loading control. Data represent mean ± SEM of three independent experiments.

**Figure 3 ijms-22-10581-f003:**
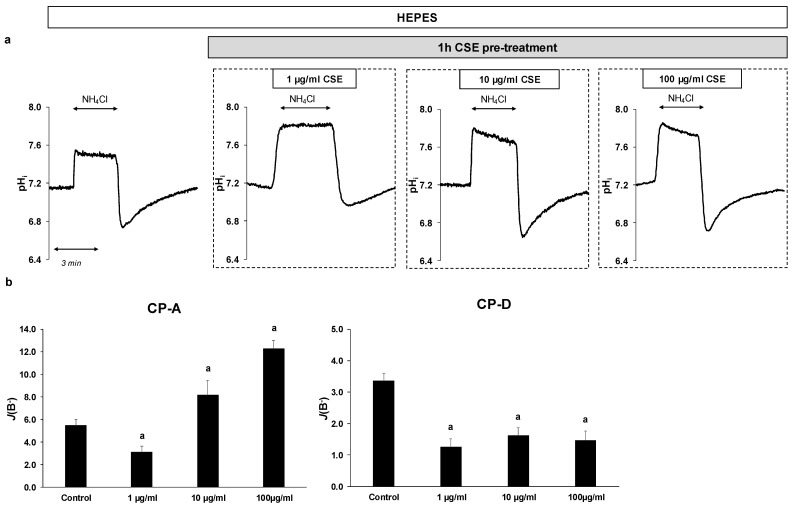
Effects of cigarette smoke extract (CSE) treatment on the activity of Na^+^/H^+^ exchanger-1 (NHE-1) in esophageal cell lines. Metaplastic (CP-A) and dysplastic (CP-D) esophageal cell lines were pretreated with different concentrations of CSE (1, 10 and 100 µg/mL) for 1 h, and the activity of NHE-1 was measured. (**a**) Representative intracellular pH (pH_i_) curves present the recovery from acidosis in CP-A cells. (**b**) Summary data of the calculated activity of NHE-1 in the different cell lines. The rate of pH recovery (*J*(B^−^)) was calculated as described in Figure 2b. Data are presented as the mean ± SEM; a = *p* ≤ 0.05 vs. control; *n* = 12–14 exp./66–68 ROIs.

**Figure 4 ijms-22-10581-f004:**
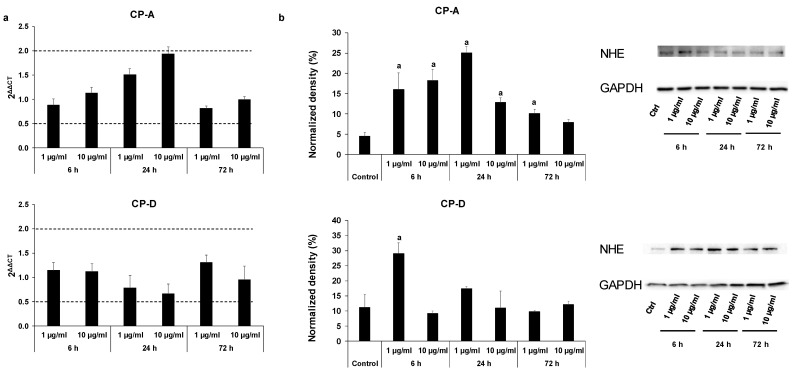
Effects of cigarette smoke extract (CSE) treatment on the mRNA and protein expression of Na^+^/H^+^ exchanger-1 (NHE-1) in esophageal cell lines. Metaplastic (CP-A) and dysplastic (CP-D) esophageal cell lines were treated with dif-ferent concentrations of CSE (1 and 10 µg/mL) for 6, 24 and 72 h, and the relative gene (**a**) and protein (**b**) expressions of NHE-1 were investigated by real-time PCR and Western blot, respectively. GAPDH was used as a protein-loading control. Data represent mean ± SEM of three independent experiments; a = *p* ≤ 0.05 vs. control.

**Figure 5 ijms-22-10581-f005:**
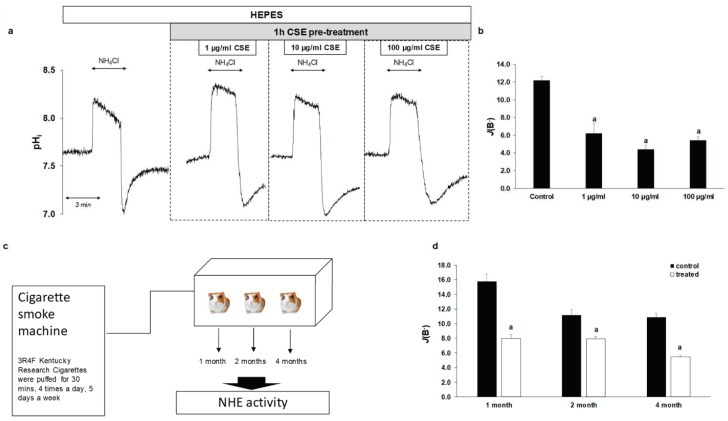
Effects of cigarette smoke extract (CSE) treatment and smoking on the activity of Na^+^/H^+^ exchanger-1 (NHE-1) in guinea pig esophageal epithelial cells (EECs). (**a**) Normal guinea pig EECs were pretreated with different concentrations of CSE (1, 10 and 100 µg/mL) for 1 h and the activity of NHE-1 was measured. Representative intracellular pH (pH_i_) curves show the recovery from acidosis. (**b**) Summary data of the calculated activity of NHE-1 in guinea pig EECs. The rate of pH recovery (*J*(B^−^)) was calculated as described in Figure 2b. Data are presented as the mean ± SEM; a: *p* ≤ 0.05 vs. control; *n* = 13–18 exp./86–99 ROIs. (**c**) Chronic effect of cigarette smoking was investigated, using smoking chamber. Guinea pigs were exposed to whole body cigarette smoke 4 times a day, 5 days a week, 30 min each time, using a TE2 closed-chamber manual smoking system. After 1, 2, or 4 months of smoking, the animals were sacrificed and NHE-1 activity was measured. (**d**) Summary data of the calculated activity of NHE-1 in guinea pig EECs. The rate of pH recovery (*J*(B^−^)) was calculated as described in Figure 2b. Data are presented as the mean ± SEM; a: *p* ≤ 0.05 vs. control; *n* = 8–16exp./81–210 ROIs.

**Figure 6 ijms-22-10581-f006:**
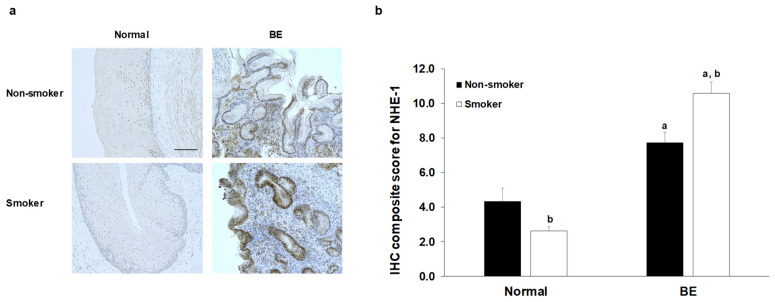
Expression of Na^+^/H^+^ exchanger-1 (NHE-1) in human esophageal samples. (**a**) Representative immunohistoche-mical stainings show the presence of NHE-1 in human esophageal samples. Scale bar represents 100 µm. (**b**) Quantification of DAB intensities were calculated, using a semi-quantitative scoring system. Data represent mean ± SEM of 23–25 specimens/3–6 patients each group; a = *p* ≤ 0.05 vs. normal; b = *p* ≤ 0.05 vs. non-smoker. BE: Barrett’s esophagus.

**Figure 7 ijms-22-10581-f007:**
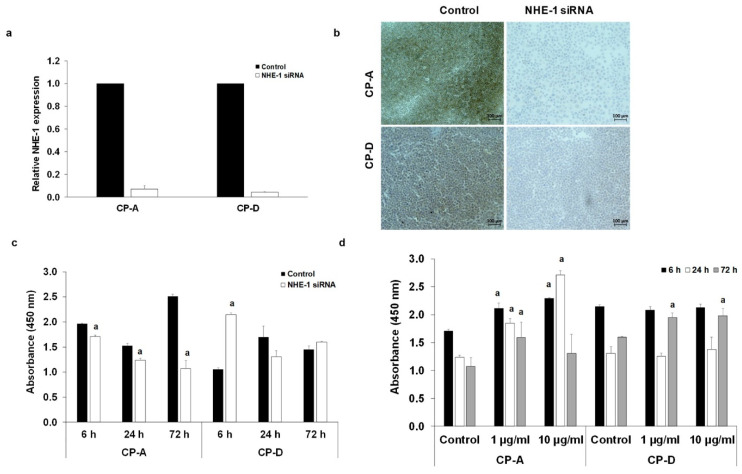
Knockdown of Na^+^/H^+^ exchanger-1 (NHE-1) in human esophageal cell lines. The expression levels of NHE-1 were investigated by RT-PCR (**a**) and immunohistochemistry (**b**) in control cells and in cells treated with specific siRNA for *SLC9A1*. The rate of proliferation was determined in the non-treated (**c**) and cigarette smoke extract-treated (**d**) CP-A and CP-D cells. Data represent mean ± SEM of three independent experiments; a = *p* ≤ 0.05 vs. control.

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
