# Peer review of "Inhibition of NHE-1 Increases Smoke-Induced Proliferative Activity of Barrett’s Esophageal Cell Line"

_ijms, 2021, doi:10.3390/ijms221910581_

Round 1

Reviewer 1 Report

The authors conducted a study hypothesized that NHE-1 is a compensatory factor protecting esophageal cells from epithelial transformation in smokers.  The experimental results of this study are well-presented and supports their hypothesis. The experimental data is concise because in vitro, in vivo and human samples are included. 

I only have a small question. NHE-1 also plays important role in against acid-induced injury in human esophagus, how would drugs including H2-blocker and proton-pump inhibitor affect NHE-1?   

Reviewer 2 Report

Please, see attached file

Reviewer 3 Report

This study reports on the role of Na+/H+ exchanger-1 (NHE-1) in smoking-induced esophageal diseases.  The role of NHE-1 was studied in metaplastic and dysplastic esophageal cell lines as well as normal esophageal cells treated with cigarette smoke extract (CSE). Protein expression of NHE-1 was investigated in normal squamous epithelium and Barrett's esophagous samples obtained from patients with smoking and non-smoking history and found to be much higher in smokers BE patients compared to normal subjects. Furthermore, silencing experiments showed that in the presence of NHE-1, CSE decreased the proliferation of metaplastic cells whereas in the absence of the exchanger, cell proliferation was increased, supporting a potential protective role of NHE-1 in BE and that low NHE-1 expression may contribute to the neoplastic progression of BE in smoking patients.

Overall the study is well-organised and well-conducted. 

Α couple of comments below:

  1. Figures 1A and 1B: No need for the decimal points in the Y axis
  2. Figure 2C: Label the Y axis in the same way as similar data appear in Figure 4A
  3. Figure 4A: It appears that there are significant differences between the treatments on the gene expression levels that may explain the protein expression levels at least in the case of the results presented in Fig.4B in CP-A cells. Was the the level of significance considered rather strict? How could the elevation at the protein level could be explained together with the increase in the activity?
  4. Figure 6: Description of  "BE"  in the figure legend is missing
  5. Future directions in the conlusive paragraph are recommended to be included. 
